# Magnetic and Magnetocaloric Properties of Nano- and Polycrystalline Manganites La_(0.7−x)_Eu_x_Ba_0.3_MnO_3_

**DOI:** 10.3390/ma15217645

**Published:** 2022-10-31

**Authors:** Roman Atanasov, Rares Bortnic, Razvan Hirian, Eniko Covaci, Tiberiu Frentiu, Florin Popa, Iosif Grigore Deac

**Affiliations:** 1Faculty of Physics, Babes-Bolyai University, Str. Kogalniceanu 1, 400084 Cluj-Napoca, Romania; 2Faculty of Chemistry and Chemical Engineering, Babes-Bolyai University, Str. Arany Janos 11, 400028 Cluj-Napoca, Romania; 3Materials Science and Engineering Department, Technical University of Cluj-Napoca, Blvd. Muncii 103-105, 400641 Cluj-Napoca, Romania

**Keywords:** manganites, nanoparticle perovskites, crystallography, magnetic behavior, phase transition, critical behavior, magnetocaloric effect

## Abstract

Here, we report synthesis and investigations of bulk and nano-sized La_(0.7−x)_Eu_x_Ba_0.3_MnO_3_ (x ≤ 0.4) compounds. The study presents a comparison between the structural and magnetic properties of the nano- and polycrystalline manganites La_(0.7−x)_Eu_x_Ba_0.3_MnO_3_, which are potential magnetocaloric materials to be used in domestic magnetic refrigeration close to room temperature. The parent compound, La_0.7_Ba_0.3_MnO_3_, has Curie temperature *T*_C_ = 340 K. The magnetocaloric effect is at its maximum around *T*_C_. To reduce this temperature below 300 K, we partially replaced the La ions with Eu ions. A solid-state reaction was used to prepare bulk polycrystalline materials, and a sol-gel method was used for the nanoparticles. X-ray diffraction was used for the structural characterization of the compounds. Transmission electron spectroscopy (TEM) evidenced nanoparticle sizes in the range of 40–80 nm. Iodometry and inductively coupled plasma optical emission spectrometry (ICP-OES) was used to investigate the oxygen content of the studied compounds. Critical exponents were calculated for all samples, with bulk samples being governed by tricritical mean field model and nanocrystalline samples governed by the 3D Heisenberg model. The bulk sample with x = 0.05 shows room temperature phase transition *T*_C_ = 297 K, which decreases with increasing x for the other samples. All nano-sized compounds show lower *T*_C_ values compared to the same bulk samples. The magnetocaloric effect in bulk samples revealed a greater magnetic entropy change in a relatively narrow temperature range, while nanoparticles show lower values, but in a temperature range several times larger. The relative cooling power for bulk and nano-sized samples exhibit approximately equal values for the same substitution level, and this fact can substantially contribute to applications in magnetic refrigeration near room temperature. By combining the magnetic properties of the nano- and polycrystalline manganites, better magnetocaloric materials can be obtained.

## 1. Introduction

The search for more efficient refrigeration methods has been ongoing ever since humans looked at the snowy peaks on the mountains from under the blistering sun [1]. Although the desire for a cooled environment and long-lasting food was ever present, no significant progress was made until the advent of electricity [2]. Then, vapor compression cooling systems became dominant as refrigeration became prevalent. However, it has been proven that such refrigeration is harmful to the environment; hence, new ways must be found and researched [2].

A good candidate for such a new method is the use of the magnetocaloric effect in, for example, intermetallic compounds of Gadolinium (Gd) [3], where the efficiency of the Carnot cycle can reach 60% [3], whereas in the conventional gas compression method (CGC), it is only about 5–10% [4]. However, since Curie temperatures of Gd alloys are lower (276 K) than that of Gd (294 K) [3], several other candidates have been investigated [5]. 

In addition, manganites of the type A_1−x_B_x_MnO_3_ (where A is a trivalent rare earth cation and B is a divalent alkaline earth cation [6] are known for their colossal magnetoresistance (CMR) effect, which is at a maximum close to the Curie temperature *T*_C_ [7]. This effect refers to when a transition from insulator to metal occurs at a temperature denoted by *T*_p_. The sharp transition from ferromagnetic to paramagnetic phase at *T*_C_ is important for a high magnetic entropy change. The two temperatures *T*_C_ and *T*_P_ are close to each other depending on the size of the domain walls; the larger the wall, the bigger the distance between them, which requires more energy to ”flip” the orientation of the neighboring domain [8]. In nanoparticles, the sizes of the particles vary and the disconnection between them causes the change in magnetic entropy to be more gradual and smaller in magnitude [9].

Besides the CMR effect, the most important property of such materials is the large magnetic entropy change which occurs when the external magnetic field varies in some compounds. In recent times, rare earth manganites such as La_1−x_Sr_x_MnO_3_ and Pr_1−x_Ba_x_MnO_3_ [10,11] have been of increasing interest for exhibiting such large entropy changes.

The optimal ratio of doping in samples (such as the parent sample for this study: La_1−x_Ba_x_MnO_3_) is x = 0.3, where the ratio of Mn^3+^ and Mn^4+^ allows for the optimal double exchange process [8]. 

Magnetic and electrical behavior of these types of compounds depend on the preparation method (which influences the domain wall size), the ratio of Mn^3+^/Mn^4+^ ions, and the size difference between the rare earth element and the alkali metal [4,7,9]. In the case where the mismatch is great, as in the case with La and Ba, a separation between *T*_C_ and *T*_P_ is observed, and in addition, the magnetic entropy change close to *T*_C_ is sharp because of strong spin-lattice coupling, which is a good sign for a high magnetocaloric effect [10]. In cases where the difference is relatively small, as with La and Ca, the grain boundary is smaller and the distance between *T*_C_ and *T*_p_ is also smaller [8]. The number of La^3+^ ions affects the critical exponents, and in the case of (La,Ba)MnO_3_, they correspond to the short-range Heisenberg model [11].

In this paper, the critical and magnetocaloric behaviors of La_(0.7−x)_Eu_x_Ba_0.3_MnO_3_ (where x = 0.05, 0.1, 0.2, 0.3, 0.4) in bulk material and nano-sized particles are discussed. La_0.7_Ba_0.3_MnO_3_ has a large magnetic entropy change at 340 K. It has been shown that substitution of Eu in place of La atoms in La_0.7_Sr_0.3_MnO_3_ samples leads to lowering of *T*_C_ [12,13] below room the temperature, where the magnetocaloric effect could be important for domestic cooling applications. As a result, the smaller ionic radius of an Eu atom was chosen for this study as a substitute for La in order to promote higher disorder and to manipulate the values of *T*_C_. Bulk compounds were prepared by solid-state reaction method, and nanoparticles were made with a modified sol-gel method. All samples have crystal structures belonging to the Rhombohedral (R-3c) symmetry group. Magnetic critical behavior analysis revealed that bulk samples are governed by the tricritical mean field model, while the nano-samples are governed by the 3D Heisenberg model. It was found that the relative cooling power increased with the level of doping, while the magnetic entropy change vs. temperature graphs were the sharpest in the sample with x = 0.05. 

The paper is organized as follows. In Section 2, we describe the preparation routes for the bulk polycrystalline and for nano-sized samples, as well as the methods we used to characterize them from structural, morphological, oxygen stoichiometric, electrical, and magnetic perspectives. In Section 3, we present the results of our investigations and the analyses of the obtained data. We also discuss the critical magnetic behavior and the magnetocaloric effect of the samples. Finally, Section 4 summarizes the conclusions resulting from this study.

## 2. Materials and Methods

The bulk samples were prepared by solid-state reaction. Precursors, consisting of oxides La_2_O_3_ (99.9%), Eu_2_O_3_ (99.99%), MnO_2_ (99.9%), and carbonate BaCO_3_ (99.9%) from Alfa Aesar, were mixed by hand in an agate mortar using a pestle for approximately 3 h each. The mixed powder was then calcinated at 1100 °C for 24 h in air. After that, the samples were pressed at 3 tons into pellets and sintered at 1350 °C for 30 h in air to produce 2 g samples.

The nano-sized samples were prepared with the sol-gel method. Nitrates of La (99.9%), Eu (99.9%), Ba (99%), and Mn (98%) from Alfa Aesar were used as precursors. They were dissolved in pure water at 60 °C for 45–60 min, after which 10 g of sucrose of 99% purity was added. The mixture was stirred for another 45 min to allow for positive ions to attach to the sucrose chain. The temperature was then reduced and pectin was added 20 min before the end of mixing in order to expand the xero-gel. The mixture was dried in a sand bath for 24 h and then placed in a high-air flow oven at 1000 °C for 2 h.

Both systems were structurally categorized using X-ray diffraction (XRD), and the data were analyzed using the FULLPROF Rietveld refinement technique. Scanning electron microscopy (SEM) was used to determine grain sizes along with the Williamson–Hall method for analyzing XRD data. EDX was used to confirm sample stoichiometry. Transmission Electron Microscopy (TEM) was used for determining the average size of nanoparticles.

Oxygen stoichiometry was determined by iodometric analytical titration and with inductively coupled plasma atomic emission spectroscopy (ICP-OES). In iodometry, an amount of the sample is placed in hydrochloric (HCl) acid, in which only Mn positive ions react with negative ions of Cl to produce Cl_2_. The gas is then pushed by nitrogen into another vessel containing potassium iodide. Iodine molecules are formed as a result, and the solution is titrated with sodium thiosulfate. Then, the ratio of Mn^3+^ and Mn^4+^ is calculated.

Electrical properties were measured using the four-point technique in a cryogen-free superconducting setup. Four-point chips, measuring voltage and current separately, were placed in a range of temperature between 10 K and 300 K in applied magnetic fields of up to 7 T.

Magnetic measurements were made using a Vibrating Sample Magnetometer (VSM) in the range of 4–300 K and in magnetic fields of up to 4 T.

## 3. Results

### 3.1. Structural Analysis

X-ray diffraction patterns show that all of the samples are single phase. The amount of impurities was smaller than 5% in all of the samples. A shift in 2θ to the right with increasing substitution of Eu indicates smaller cell dimensions. The patterns for nanocrystalline samples exhibit wider peaks due to their sizes. Figure 1 shows stacked XRD patterns for bulk and nanocrystalline samples, respectively.

Rietveld refinement analysis confirms the rhombohedral structure of the parent compound for each studied sample with an R-3c space group. Figure 2 presents the values for lattice parameter (*a*) and cell volume (V) for all of the samples as a function of Eu content (x). As observed, with increasing substitution level, the lattice parameter becomes smaller. This is due to Eu^3+^ ions having a smaller ionic radius (1.206 Å) than La^3+^ ions (1.5 Å) [14,15]. In turn, this changes the Mn-O-Mn angle, creating distortion in the Mn-O octahedral [14].

In order to understand the stability of these structures, the Goldschmidt tolerance factor was calculated using following relation [7,14]: (1)$$t=\frac{R_{A}+R_{0}}{\sqrt{2}\left(R_{B}+R_{0}\right)},$$
where R_A_ is the radius of A cation, R_B_ is the radius of B cation, and R_0_ is the radius of the anion.

It should be noted that with an increasing Eu ion content in the samples, the tolerance factor decreases slightly but remains consistent with keeping an orthorhombic/rhombohedral structure. Eu ions cause a decrease in R_A_ and an increase in disorder [14]. In turn, this will decrease orbital overlap and the band gap. The values of the tolerance factor are within the values for an orthorhombic/rhombohedral structure [15]. The angle of Mn-O-Mn bonds increased in the parent samples for both bulk and nanocrystalline samples from 167° to 169° for x = 0.05. Furthermore, as shown in Table 1 and Table 2, the bond length of Mn-O diminishes with each additional substitution.

The Williamson–Hall (W-H) method [16] for determining crystallite sizes was used for both systems. Table 1 shows the calculation for nanocrystalline samples with an average size range of 30–55 nm for the crystallites. This agrees with the results of TEM investigation. Pictured in Figure 3, TEM shows that the average size of the particles is about 50 nm, varying between 30 nm and 70 nm. Widely used Scherrer size calculations do not take into account the strain between the grains, and thus, they tend to be lower in value. According to Williamson–Hall calculations, the size varies from 106 nm to 172 nm, while scanning electron microscopy (SEM) shows the grain size to be 3–10 μm. The same can be observed for Rietveld crystallite size results; although they are bigger than from the WH method, they are still smaller than the SEM results. This can be attributed to the fact that a single grain contains several crystallites.

Energy dispersive X-ray spectroscopy (EDX) was also carried out for the bulk samples. As seen in Figure 4b, where a typical example is presented, the stoichiometry of heavy elements (including lanthanum, europium, barium, and manganese) is in very good agreement with the theoretical values. The distribution of the elements on the surface is considerably uniform. It is good to note here that oxygen is a much smaller atom and does not interact with X-rays nor heavier atoms [17] (pp. 279–307). For that reason, iodometry was implemented as a reliable way to calculate oxygen content.

### 3.2. Oxygen Content

Iodometric titration and inductively coupled plasma atomic emission spectroscopy were used to study the oxygen content of the bulk samples.

Iodometry is a reliable and popular method of determining oxygen content in manganites [18], as it involves direct measurement of the fraction of Mn^3+^ vs. Mn^4+^ ions. In this study, all of the samples exhibited an excess of Mn^3+^ content. This could be attributed to the oxygen deficiency during calcination and sintering. An average of 75% Mn^3+^ would result in an average oxygen content of O_2.97_ in the range of 2.96–2.99 [19], which would affect its electrical and magnetic properties [19]. The experiment showed acceptable dispersion and error. Standard deviation is a measure of dispersion of data values, or how close they are to the “mean” value, while relative standard deviation is the percentage value of the standard deviation around the “mean”. In our study, the relative standard deviation did not exceed 2.7%, showing close proximity to the mean. Results are shown in Table 3.

The results of iodometry were confirmed with inductively coupled plasma optical emission spectroscopy (ICP-OES) [20]. The process involves passing of the elements through a plasma of argon, which causes excitation and emission of specific wavelengths of light. The results are presented in Table 3. The average oxygen content according to ICP-OES falls well within the error limit of the iodometry results.

### 3.3. Electrical Measurements

An investigation of electrical resistivity at 0, 1, 2 T was carried out using the four-point probe method. The graphs are shown in Figure 5. As can be seen, the sample with x = 0.3 exhibits an expected behavior typical of a ferromagnetic manganite, with a maximum at *T*_p_. The samples with x = 0–0.2 have similar behaviors, while the sample with x = 0.4 exhibits semi-conducting behavior. The results for Curie temperatures (*T*_C_, as obtained from magnetic measurements), *T*_p_, the values of the resistivities in the absence of a magnetic field (ρ_peak_, at 0 T), and the values of magnetoresitance (*MR*) at *T*_p_ are presented in Table 4. Magnetoresistance was calculated using the following formula [16]:*MR*% = [(ρ(H) − ρ(0))/ρ(0)] × 100,(2)

The first observation to be made here is that the *T*_p_ metallic–insulator transition temperature for each sample at 0 T magnetic field is lower than its *T*_C_; for example, *T*_C_ (x = 0.05) = 297 K, and *T*_p_ (x = 0.05) = 256 K. This is due to the effect of grain boundaries which act as a semiconductor (or insulator) pushing the inter-grain coupling to lower temperatures [21]. When an external magnetic field is applied, the peak shifts to the right, increasing the conductive properties of the sample. This can be attributed to the lowering of spin fluctuations and to the delocalization of charge carriers caused by the applied magnetic field, which improves the double exchange interaction [22].

Each addition of a smaller-sized Eu^3+^ ion in place of an La^3+^ ion causes disorder [7]. It also changes the angle between Mn-O-Mn by “pulling” oxygen towards the A-site [23]. A decrease in the angle changes the overlap of the electron orbital, which reduces the hopping amplitude of the electrons and causes them to be more localized. This can be observed in the systematic lowering of the *T*_p_ of samples with an increasing level of substitution. For high Eu content (x = 0.4), the disorder and the reduced value of the Mn-O-Mn angle resulted in a semiconductor-like behavior of the electrical conductivity of the compound.

From the values for resistivity (ρ_peak_) and magnetoresistance (*MR*_Max_) in Table 4, it is eviden, that both of them tend to increase with increasing substitution level of Eu. The maximum observed resistivity for the x = 0.3 sample is 240 Ω٠cm and (*MR*_Max_) (2 T) = 63.6%, while the resistivity increases to several orders of MΩ·m for x = 0.4.

The samples with x < 0.4 have typical CMR resistivity behavior as a function of temperature and applied magnetic field, as can be seen in Figure 5, except in the low-temperature region where an upturn of resistivity occurs. The analysis of these temperature dependences is usually made both for the metallic regime before metal–insulator transition (MIT) and for the semiconducting behavior in the range of higher temperatures [24]. Within the metallic region behavior, the dominant scattering phenomena are electron-electron and electron–magnon [24] (pp. 21–32) with ρ = ρ_0_ + ρ_2_*T*^2^ + ρ_4.5_*T*^4.5^. At higher temperatures, after MIT, the resistivity shows semiconductor behavior and its temperature dependence can be described by using the variable range hopping (VRH) and small polaron hopping (SPH) models [25].

For the semiconducting sample, with x = 0.4 (Figure 5b), the best fit for the resistivity behavior is the expression corresponding to the VRH model for a three-dimensional system: ρ(*T*) = ρ_0_ exp (*T*_0_/*T*)^0.25^ (as shown in the inset of Figure 5b, for μ_0_*H* = 0 T), where ρ_0_ is the prefactor and *T*_0_ is a characteristic temperature which is related to the density of states at the Fermi level and to the localization length.

It is interesting that in spite of the semiconducting behavior, this sample shows CMR properties. This behavior suggests an electrical conduction mechanism which takes place (by tunneling) between isolated manganite grains which have negative magnetoresistance.

The behavior of resistivity at temperatures below *T*_p_ is of interest in this study. A minimum in resistivity can be observed at around 30–50 K before resistivity increases again. This behavior is exhibited by all samples except the one with the highest amount of Eu, where resistivity increases drastically below *T*_p_. In the literature, the minimum in resistivity was partially attributed to Kondo-like effects [14]. These minima are caused by small magnetic impurities which localize electrons of opposite spin, thus increasing the scattering of conduction electrons. However, this scenario is quite different from that of polycrystalline manganites with different size grains separated by (disordered matter) grain boundaries; this rules out the hypothesis of the Kondo effect [26].

The upturn can be better explained by the combined effect of electron–electron interactions, electron–phonon scattering, and weak localization [24,27]. In addition, the disorder and strain from the grain boundaries can also act as supplementary localization factors of charge carriers. Besides these, the electrical conductivity of the grain boundaries depreciates with decreasing temperature, leading to increased resistivity. The upturn in the thermal dependence of resistivity is a consequence of both intrinsic (intragrain) effects and extrinsic grain boundary scattering/tunneling effects [28], as was also found in some other polycrystalline complex transition metal oxides. The sample with the highest Eu content exhibits a continual increase in resistivity below *T*_p_, which can be explained by the size and quality of the grain boundaries. It is evident that grain boundaries play a dominant role in the electrical behavior of the samples.

### 3.4. Magnetic Properties

All of the samples exhibit ferromagnetic–paramagnetic transitions. Typical and selected magnetization vs. temperature (*M*(*T*)) plots are presented in Figure 6. Curie temperatures can be calculated from the derivative of the magnetization with respect the temperature, with the inflection point corresponding to *T*_c_, which can be seen in Table 5, Table 6 and Table 7 [29]. With increasing Eu content x, the values of *T*_c_ gradually and systematically become lower as a result of the induced disorder. The sample with x = 0.05 has Curie temperature *T*_C_ = 297 K.

Nano-scale particles also show ferromagnetic behavior, but their *T*_C_ values are lower compared to the equivalent bulk material, as shown in Figure 6. This is due to the size of particles and their ”surface effects” which occur as a result of a large surface-to-volume ratio [30], i.e., disorder effects in the surface layer of the particles which contain an increased number of broken chemical bond and other defects, resulting in spin canting and reducing the magnetic moment of the particles [31]. An average difference in *T*_C_ values is 70–90 K for the samples with the same substitution level: *T*_C_ (x = 0.1 bulk) = 270 K and *T*_C_ (x = 0.1 nano) = 200 K. It is also interesting that the slope of the magnetization change of the *M*(*T*) curves in nanoparticles is much lower than in bulk, which can be attributed to the distribution of the particle sizes within the samples. In general, bulk material has higher values of magnetization and a narrower temperature range of the magnetic phase transition, as can be seen in Figure 6.

Arrott plots (*M*^2^ vs. *H*/*M*) allow the determination of the magnetic phase transition order [32]. According to the Banerjee criterion, positive and negative slopes of the curves correspond to second- and first-order magnetic phase transitions, respectively [28]. All of the samples exhibit positive slopes for these curves, indicating second-order magnetic phase transitions. A selected sample of Arrott plots are presented in Figure 7.

Arrott plots are based on Landau’s mean field theory, [33]. The Gibbs free energy around a critical point is defined as
*G* (*T,M*) = *G*_O_ + *MH* + *aM*^2^ + *bM*^4^ +…,(3)
where *a* and *b* are coefficients which depend on temperature. Minimizing the Gibbs free energy with respect to magnetization, we obtain
*H/M* = 2*a* + 4*bM*^2^,(4)

According to Gibbs free energy equations, the isotherm lines must be parallel and straight, but that is not observed in Arrott plots. The problem lies in inexactness of the critical exponents [34]. β relates to spontaneous magnetization and α is related to the inverse of susceptibility χ [34,35].

These equations can be generalized as follows [34]:*M*_S_(*T*) = *M*_0_ (−*ε*)*^β^*, *T* < *T*_C_,(5)
(6)$$\chi^{-1}\left(T\right)=\left(\frac{h_{0}}{M_{0}}\right)\varepsilon^{\gamma },\;T\;>\;T_{C},$$
*M**= D* (μ_0_*H*^1*/**δ*^), *T* = *T*_C_,(7)
where *ε* is the reduced temperature (*T* − *T*_c_)/*T*_c_ and *M*_0_, *h*_0_/*M*_0_, and *D* are critical amplitudes.

For mean field theory, we have γ = 1, β = 0.5, and δ = 3. For the 3D Heisenberg model, we have γ = 1.366, β = 0.355, and δ = 4.8. For the Ising model, we have γ = 1.24, β = 0.325, and δ = 4.82. For the tricritical mean field model, we have γ = 1, β = 0.25, and δ = 5 [24,36].

The modified Arrott plot method [35] is an iterative method. It begins with an Arrott–Noakes plot (*M*^1/β^ vs. μ_0_*H*/*M*^1/γ^), which involves finding proper exponents which will make the lines parallel and straight to determine the β and γ critical exponents [10]. Spontaneous magnetization *M*_S_(T,0) is found from the intercepts of isotherms with the ordinate of the plot. The inverse of the susceptibility χ_0_^−1^(T) is taken from the intercept with the abscissa. Further fitting of these values into Equations (6)–(8) refines the values to those very close to real ones. The Widom scaling relation β + γ = β δ [10] gives the value of δ. Selected graphs for modified Arrott plots are presented in Figure 8; the data are presented in full in Table 5.

**Table 5 materials-15-07645-t005:** Critical exponent values for all samples.

Compound		γ	β	δ	*T*_c_ (K)
x= 0	bulk	1.065	0.288	4.69	340
x = 0.05	bulk	0.915	0.234	4.9	297
x = 0.1	bulk	1.07	0.24	5.45	270
x = 0.2	bulk	0.976	0.246	4.967	198
x = 0.3	bulk	0.933	0.255	4.659	142
x= 0.4	bulk	1.022	0.249	5.104	99
x = 0	nano	1.823	0.493	4.698	263
x= 0.05	nano	1.968	0.548	4.589	220
x = 0.1	nano	1.867	0.521	4.584	200
x = 0.2	nano	1.755	0.477	4.679	136
x= 0.3	nano	1.931	0.537	4.596	90
x = 0.4	nano	1.789	0.512	4.49	64
Mean field model		1	0.5	3	
3D Heisenberg model		1.366	0.355	4.8	
Ising model		1.24	0.325	4.82	
Tricritical mean field model		1	0.25	5	

Figure 9 shows the values for critical exponents for two of the samples: one of bulk and one of nano-scale. It is evident that the bulk samples are more closely governed by the tricritical mean field model (β = 0.234, γ = 0.915, δ = 4.9) for ferromagnets and the nanoparticle samples are governed by the 3D Heisenberg model (β = 0.548, γ = 1.968, δ = 4.589), rather than mean field theory model.

Usually, the 3D Heisenberg model can describe the critical properties of the short-range interactions in doped manganites, together with other theoretical models, such as the mean field and tricritical mean field models. The critical exponent values are related to the range of exchange interaction *J*(r), spin, and system dimensionality. Within renormalization group theory [37], *J*(*r*) = 1/r*^d+σ^* (*d*—dimensionality of the system; *σ*—range of interaction). For *σ* greater than 2, the 3D Heisenberg model is valid. For *σ <* 3/2, the mean field theory of long-range interaction is valid. For an intermediate range, a different universality class occurs. For the tricritical point, the critical exponents are universal: β = 0.25, γ = 1, and δ = 5. The tricritical point sets a boundary between two different ranges of order phase transitions. In this study, we focused on the general different magnetic properties of nano- and bulk polycrystalline manganites, not on their high-detail magnetic critical behavior. The modified Arrott plot (MAP) method is very accurate, and its agreement with the Kouvel–Fisher (KF) method and critical isotherm (CI) plots is usually quite remarkable [24,38]; indeed, some authors only analyze these data [39]. This is why we report here the results obtained by using MAP analysis.

All of the samples exhibit a very small coercive field as evident from hysteresis curves, with the largest values of 170 Oe for x = 0.05 bulk and 960 Oe for x = 0.4 nano-sample. This can be largely attributed to low anisotropy and lack of pinning sites [40]. Nanoparticles show greater coercivity than their bulk counterparts, as shown in Table 6 and Table 7. It has been found, in previous works, that nanoparticle coercivity tends to increase with a decreasing size in the multi-domain range, and then it decreases in the single-domain range until it reaches a superparamagnetic state, when it becomes zero [41]. Low coercivity is of utmost importance for the magnetocaloric effect.

Suitability for cooling via the magnetocaloric effect is determined by the value of magnetic entropy change in the material [40]. An appropriate ratio between Mn^3+^ and Mn^4+^ is needed for optimal ferromagnetic behavior. About 30% Mn^4+^ gives the best results [8]. When the number of charge carries increases, the risk of entering a charge ordered state occurs, where electron hopping is prohibited by rigid atomic distribution [8]. For second-order phase transitions (which were established via Arrott plots earlier), magnetic entropy change (Δ*S*_M_) can be calculated from magnetization *M* (μ_0_*H*) isotherm data and is approximated by the following equation [42]:(8)$$\Delta S_{m}\left(T,H_{0}\right)=S_{m}\left(T,H_{0}\right)-S_{m}\left(T,0\right)=\frac{1}{\Delta T}\int_{0}^{H_{0}}\;\left[M\left(T+\Delta T,H\right)-M\left(T,H\right)\right]dH,\;$$

To estimate the magnetocaloric effect, we plot −Δ*S_M_* vs. *T* (temperature) for values of external magnetic field (μ_0_*H*) of 1, 2, 3, 4 T in Figure 10. Furthermore, relative cooling power (*RCP)* is calculated as the product of entropy change (Δ*S_M_*) and temperature change at half maximum (*δ**T_FWHM_*)
(9)$$RCP\left(S\right)=-\Delta S_{m}\left(T,H\right)\times \delta T_{FWHM}$$

Table 6 shows the progression of magnetic entropy change in bulk samples and Table 7 shows the same for nano-sized samples. It is evident that maximum magnetic entropy change occurs at temperatures very close to *T*_c_. Of interest in both systems is the fact that both bulk and nanocrystalline samples with x ≤ 0.3 exhibit *RCP* values which are in the range of the values recommended for magnetocaloric materials [43]. An interesting note to be made is that the values of Relative Cooling Power (*RCP*) for similar samples are comparable to each other. *RCP* (x = 0.2) bulk = 212.4 9 (J/kg) and *RCP* (x = 0.2) nano = 218.6 (J/kg) (Table 6 and Table 7) for μ_0_Δ*H* = 4 T. The values for the maximum magnetic entropy change are very good in comparison with similar compounds presented in the literature as potential magnetocaloric materials [10,44]. Another interesting and useful note is the shape of the curves of magnetic entropy change for bulk and nanocrystalline samples. The bulk material exhibits a sharp curve, while nanoparticles have lower but wider curves. This is due to the size distribution of the nanoparticles and their separation. While the peak entropy change |Δ*S*_M_| for bulk material is higher (4.2 J/KgK) than for nano-material (1.63 J/kgK), the width of the temperature change (δ*T*_FWHM_) is reversed, with 37 K for bulk and 95 K for equivalent samples (x = 0.05) when μ_0_Δ*H* = 4 T. This fact can be useful for the cooling industry, where a wider range of applications is preferable.

**Table 6 materials-15-07645-t006:** Experimental values for La_0.7−_xEu_x_Ba_0.3_MnO_3_ bulk materials: magnetic measurements.

Compound (Bulk)	*T*_C_ (K)	*M_s_* (μ_B_/f.u.)	*H*_ci_ (Oe)	|Δ*S*_M_| (J/kgK) μ_0_Δ*H* = 1 T	|Δ*S*_M_| (J/kgK) μ_0_Δ*H* = 4 T	*RCP* (*S*) (J/kgK) μ_0_Δ*H* = 1 T	*RCP* (*S*) (J/kgK) μ_0_Δ*H* = 4 T	*Refs*
La_0.7_Ba_0.3_MnO_3_	340	4.04	200	1.33	3.5	53.7	158.4	This work
La_0.65_Eu_0.05_Ba_0.3_MnO_3_	297	3.87	172	1.71	4.2	42.7	155.4	This work
La_0.6_Eu_0.1_Ba_0.3_MnO_3_	270	3.84	63	1.6	4.1	40	187.7	This work
La_0.5_Eu_0.2_Ba_0.3_MnO_3_	198	3.7	67	1.41	3.7	38.1	212.6	This work
La_0.4_Eu_0.3_Ba_0.3_MnO_3_	142	3.78	66	1.7	3.5	42.6	176.4	This work
La_0.3_Eu_0.4_Ba_0.3_MnO_3_	99	3.46	120	1.02	2.83	25.7	133.3	This work
La_0.7_Ca_0.3_MnO_3_	256			1.38		41		[10]
La_0.7_Sr_0.3_MnO_3_	365			-	4.44 (5 T)		128 (5 T)	[10]
La_0.6_Nd_0.1_Ca_0.3_MnO_3_	233			1.95		37		[10]
Gd_5_Si_2_Ge_2_	276			-	18 (5 T)	-	535 (5 T)	[10]
Gd	293			2.8		35		[10]

**Table 7 materials-15-07645-t007:** Experimental values for La_0.7−x_Eu_x_Ba_0.3_MnO_3_ nano materials.

Compound (Nano)	*T*c (K)	*M_s_* (μ_B_/f.u.)	*H*_ci_ (Oe)	|Δ*S*_M_| (J/kgK) μ_0_Δ*H* = 1 T	|Δ*S*_M_| (J/kgK) μ_0_Δ*H* = 4 T	*RCP*(*S*) (J/kgK) μ_0_Δ*H* = 1 T	*RCP*(*S*) (J/kgK) μ_0_Δ*H* = 1 T	*Refs*
La_0.7_Ba_0.3_MnO_3_	263	2.79	4800	1.04	1.37	105.4	130.1	This work
La_0.65_Eu_0.05_Ba_0.3_MnO_3_	220	2.95	410	0.43	1.63	43.3	155.6	This work
La_0.6_Eu_0.1_Ba_0.3_MnO_3_	200	2.6	390	0.93	1.23	93.5	135.3	This work
La_0.5_Eu_0.2_Ba_0.3_MnO_3_	136	2.96	280	0.46	1.68	47.8	218.4	This work
La_0.4_Eu_0.3_Ba_0.3_MnO_3_	90	2.3	590	0.39	1.99	38.3	187.7	This work
La_0.3_Eu_0.4_Ba_0.3_MnO_3_	64	2.09	960	0.25	1.09	23.3	119.9	This work
La_0.67_Ca_0.33_MnO_3_	260				0.97 (5 T)		27 (5 T)	[45]
Pr_0.65_(Ca_0.6_Sr_0.4_)_0.35_MnO_3_	220			0.75		21.8		[46]
La_0.6_Sr_0.4_MnO_3_	365			1.5		66		[47]

## 4. Conclusions

The compounds La_0.7−x_Eu_x_Ba_0.3_MnO_3_ (x = 0, 0.05, 0.1, 0.2, 0.3, 0.4) were synthetized by solid-state reaction to produce a bulk material whose crystalline structure and morphology were investigated by XRD and SEM. The sol-gel method was used to produce nano-scale particles, which were analyzed by XRD and TEM, showing an average size of 30–70 nm. Both systems are single phase, with rhombohedral (R-3c) lattice symmetry. Cell parameters diminish with addition of Eu in the lattice structure. Mn-O bond length tends to shorten with an increasing level of Eu, and the addition of Eu increases the angle Mn-O-Mn. ZFC-FC plots suggest single magnetic phase and low magnetic anisotropy. The sample with x = 0.05 shows a magnetic phase transition at 297 K for bulk compound, while the La_0.7_Ba_0.3_MnO_3_ nanoparticle sample has *T*_C_ = 263 K. Both systems show very small coercivity, with nano-sized samples being slightly larger: 63 Oe for the bulk sample with x = 0.1 and 390 Oe for the corresponding nanocrystalline sample. Iodometry was used in order to estimate oxygen content in samples, showing a lower concentration of Mn^4+^ ions, leading to the lowest value of oxygen content to be O_2.97±0.02_ for the x = 0.4 sample. General oxygen deficiency was confirmed by inductively coupled plasma optical emission spectrometry (ICP-OES). The samples with x < 0.4 show metallic–insulator transition at a temperature *T*_p_ lower than magnetic transition temperatures *T*_c_. The sample with x = 0.4 exhibits increasing resistivity with decreasing temperature below 160 K, suggesting either the ferromagnetic clusters are non-metallic or they are too small to compensate the increase of resistivity induced by disorder. All samples show negative magnetoresistance. The sample with x = 0.3 exhibits magnetoresistance *MR* (2 T) = 63.6%. Arrott plots confirm second-order magnetic phase transition for all of the samples. Modified Arrott plot analysis revealed the critical exponents for bulk samples to be in the tricritical mean field model range and in the 3D Heisenberg model range for nanocrystalline samples. The maximum magnetic entropy change of 4.2 J/kgK was observed for the x = 0.05 bulk sample for µ_0_Δ*H* = 4 T. Nanocrystalline samples exhibit lower peak magnetic entropy change (1.63 J/kgK for x = 0.05), but *T*_fwhm_ exceeds 100 K (130K for x = 0.2). Relative cooling power *RCP* is comparable between equivalent samples for both systems: 212.4 J/kg for x = 0.3 in nanocrystalline samples and 212.6 J/kg for bulk. The value of *RCP* close to room temperature phase transition *T*_C_ = 297 K for the bulk sample La_0.65_Eu_0.05_Ba_0.3_MnO_3_ is 155.4 J/Kg, possessing the required parameters of a magnetocaloric material. Since the temperature range (δ*T*_FWHM_) for nano-sized samples La_0.7_Ba_0.3_MnO_3_ and La_0.65_Eu_0.05_Ba_0.3_MnO_3_ covers a wide range (95 K) including room temperature, they may be used in multistep refrigeration processes. However, we should be cautious in taking high RCP values for nano-materials at face value [48]; potentially, fewer compounds would be necessary for use in stacks of refrigeration systems. This wide range of effective cooling in nanoparticles together with high entropy change in bulk material can be combined for suitable commercial cooling.

## Figures and Tables

**Figure 1 materials-15-07645-f001:**
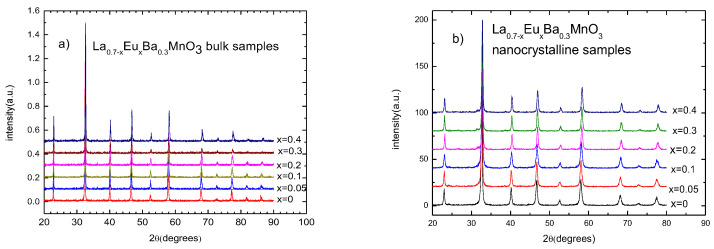
X-ray diffraction patterns for (**a**) La_0.7−x_Eu_x_MnO_3_ polycrystalline bulk samples and (**b**) La_0.7−x_Eu_x_MnO_3_ nano-sized samples.

**Figure 2 materials-15-07645-f002:**
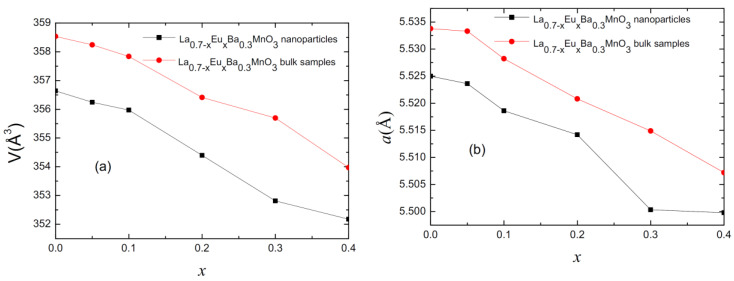
(**a**) Plot of cell volume for polycrystalline bulk and nano-sized samples. (**b**) Plot of cell parameter “*a*” for polycrystalline bulk and nano-sized samples.

**Figure 3 materials-15-07645-f003:**
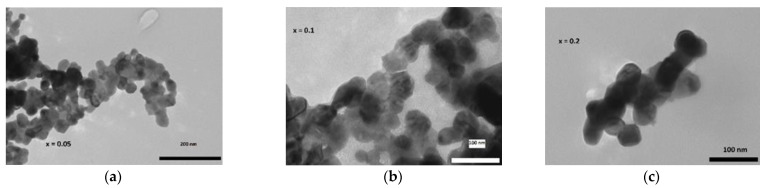
Selected TEM pictures for La_1−x_Eu_x_MnO_3_ nano-sized samples for x = 0.05 (**a**), x = 0.1 (**b**), x = 0.2 (**c**).

**Figure 4 materials-15-07645-f004:**
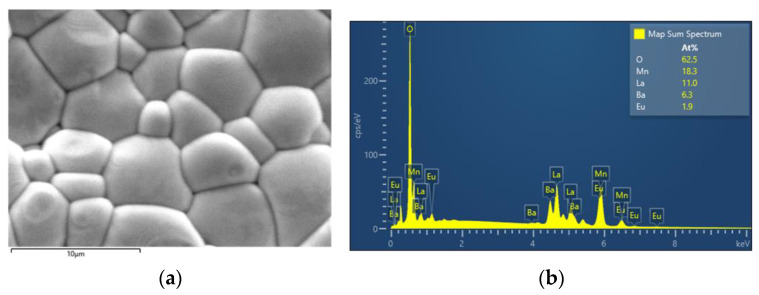
(**a**) SEM picture of the La_0.65_Eu_0.05_Ba_0.3_MnO_3_ sample. (**b**) EDX for the La_0.65_Eu_0.05_Ba_0.3_MnO_3_ sample.

**Figure 5 materials-15-07645-f005:**
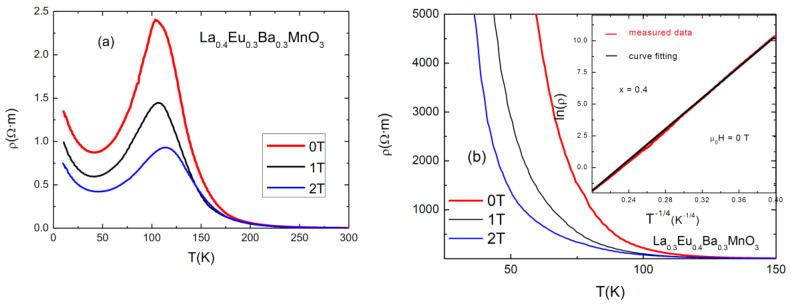
Resistivity vs. temperature graphs for (**a**) x = 0.3 and (**b**) x = 0.4. The inset shows the fitting of ln(ρ) as a function of T^−1/4^ for μ_0_*H* = 0 T.

**Figure 6 materials-15-07645-f006:**
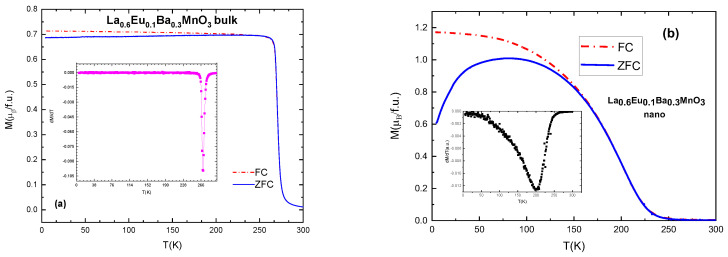
(**a**) ZFC-FC curves and derivative of magnetization (in inset) for the bulk sample with x = 0.1; (**b**) ZFC-FC curves and derivative (in inset) for nanocrystalline sample with x = 0.1.

**Figure 7 materials-15-07645-f007:**
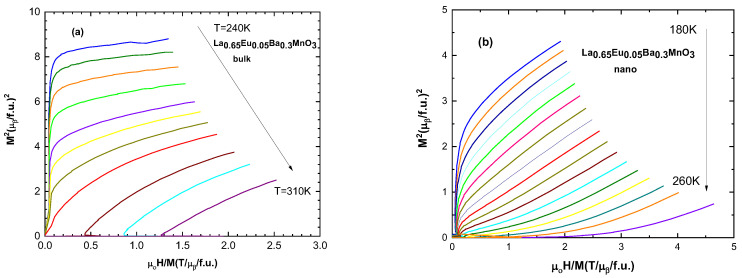
Arrott plot (*M*^2^ vs. *H/M*) for (**a**) the bulk sample with x = 0.05 and for (**b**) the nanocrystalline sample with x = 0.05.

**Figure 8 materials-15-07645-f008:**
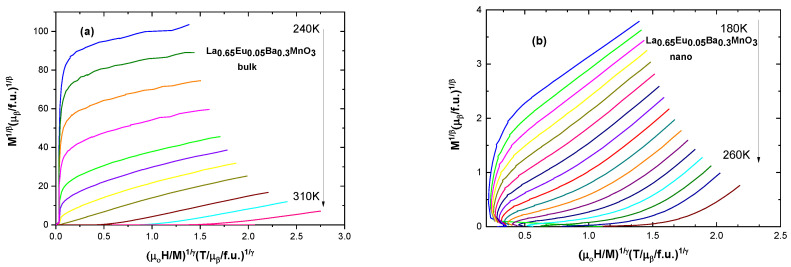
Modified Arrott plots for (**a**) the bulk sample with x = 0.05 and for (**b**) the nanocrystalline sample with x = 0.05.

**Figure 9 materials-15-07645-f009:**
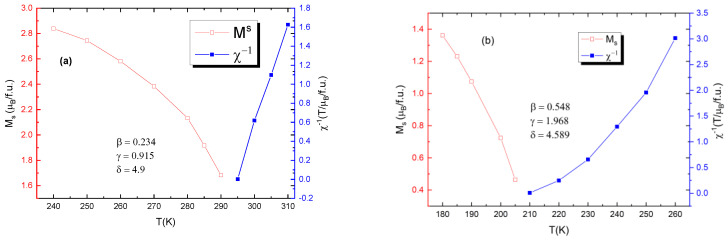
Calculated values for critical exponents for (**a**) the bulk sample with x = 0.05 (**b**) for the nanocrystalline sample with x = 0.05.

**Figure 10 materials-15-07645-f010:**
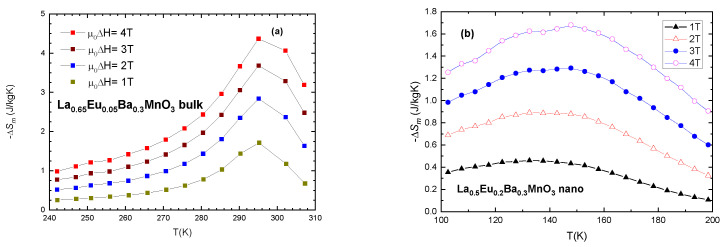
Magnetic entropy change vs. temperature for selected samples: (**a**) x = 0.05 bulk and (**b**) x = 0.2 nanocrystalline sample.

**Table 1 materials-15-07645-t001:** Calculated tolerance factors, Mn-O lengths, and crystallite sizes for nanocrystalline samples using the Williamson–Hall and Rietveld methods, including strain values.

Eu Content (Nano)	*t* (Tolerance Factor)	Mn-O (Å)	Williamson–Hall Size (nm)	Average Rietveld Size (nm)	Strain
x = 0	0.997	1.962	36.23	18.14	0.0023
x = 0.05	0.992	1.959	41.46	20.62	0.0024
x = 0.1	0.987	1.958	29.15	21.27	0.0019
x = 0.2	0.976	1.956	54.61	29.03	0.0023
x = 0.3	0.966	1.953	49.95	31.57	0.0019
x = 0.4	0.956	1.952	45.74	34.37	0.0017

**Table 2 materials-15-07645-t002:** Calculated Mn-O lengths and crystallite sizes for polycrystalline bulk samples using the Williamson–Hall and Rietveld methods, including strain values.

Eu Content (Bulk)	Williamson–Hall Size (nm)	Mn-O (Å)	Average Rietveld Size (nm)	Strain
**x = 0**	110.05	1.966	111.1	0.0017
**x = 0.05**	110.04	1.963	435.97	0.0018
**x = 0.1**	172.03	1.962	238.22	0.0017
**x = 0.2**	146.36	1.959	324.41	0.002
**x = 0.3**	128.38	1.958	246.71	0.0016
**x = 0.4**	106.05	1.955	144.42	0.0016

**Table 3 materials-15-07645-t003:** Average oxygen content calculated using iodometry and inductively coupled plasma optical emission spectrometry (ICP-OES).

Eu Content	Average Mn^3+^ Ratio	Standard Deviation	Relative Standard Deviation (%)	Average Oxygen Content	ICP-OES
x = 0	0.7306	0.0159	2.18	O_2.98±0.02_	O_2.94±0.14_
x = 0.05	0.7257	0.0083	1.14	O_2.99±0.01_	O_2.93±0.13_
x = 0.1	0.7032	0.0189	2.69	O_2.99±0.02_	O_2.99±0.13_
x = 0.2	0.7282	0.0136	1.87	O_2.98±0.02_	O_3.13±0.15_
x = 0.3	0.7565	0.0076	0.99	O_2.97±0.01_	O_3.03±0.15_
x = 0.4	0.7612	0.0138	1.81	O_2.97±0.02_	O_3.13±0.17_

**Table 4 materials-15-07645-t004:** Experimental values for La_0.7−x_Eu_x_Ba_0.3_MnO_3_ bulk materials: electrical properties.

Compound (Bulk)	*T*_C_ (K)	*T*_P_ (K)	ρ_peak_ (Ωcm) in 0 T	*MR*_Max_ (%) (1 T)	*MR*_Max_ (%) (2 T)
La_0.7_Ba_0.3_MnO_3_	340	295	0.693	5.8	12.9
La_0.65_Eu_0.05_Ba_0.3_MnO_3_	297	256	0.812	4.2	11.8
La_0.6_Eu_0.1_Ba_0.3_MnO_3_	270	220	0.084	32.9	52.6
La_0.5_Eu_0.2_Ba_0.3_MnO_3_	198	165	21.753	22.7	42.1
La_0.4_Eu_0.3_Ba_0.3_MnO_3_	142	103	240.455	40.4	63.6
La_0.3_Eu_0.4_Ba_0.3_MnO_3_	99	-	100 × 10^9^	-	-

## Data Availability

Data presented in this study are available in this article.

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
