# Peer review of "Magnetic and Magnetocaloric Properties of Nano- and Polycrystalline Manganites La(0.7−x)EuxBa0.3MnO3"

_materials, 2022, doi:10.3390/ma15217645_

Round 1
Reviewer 1 Report
The authors studied the magnetocaloric behavior of La(0.7-x)EuxBa0.3MnO3 in bulk material and nano size particles. nano-scale particles were produced by Sol-gel method and were analyzed by X-ray diffraction and Transmission Electron Microscopy. The manuscript is well-written and also interesting. I recommend it for publication with some minor comments.
1- Please check the logic of the abstract. Please add sentences to explain the meaning, the main points, the improvement, and the promising application of the study. Plenty of detailed data have been given, however, in the abstract, important procedures and results should be mentioned in a simple manner. What is the novelty of the work and where does it go beyond previous efforts in the literature?
2- At the end of the first section, there is no information on how the article is organized.
3- The punctuation marks need to be administered after each equation. The punctuation marks should be improved. All notations in the entire manuscript should be explained.
Reviewer 2 Report
Article " Magnetic and magnetocaloric properties of nano- and polycrystalline manganites La(0.7-x)EuxBa0.3MnO3" Authors: Roman Atanasov , Rares Bortnic , Razvan Hirian , Eniko Covaci , Tiberiu Frentiu , Florin Popa , Iosif Grigore Deac are dedicated to topical topics of the modern world: creation of materials for magnetocaloric refrigeration systems. The article corresponds to the subject of the journal " Materials", Section Advanced Nanomaterials and Nanotechnology. The article is well structured, written in a clear and understandable language, the conclusions are logical, the literature corresponds to the stated topic. However, there are some notes:
1) The authors use the phrase "investigations of bulk and nano-sized" compounds. Everything is good in this sentence. Further, the authors use the phrase "bulk samples" and "nano-samples" throughout the text of the article. The phrase "nano-samples" is incorrect or is a controversial term. It will be correct to write a sample of nanoparticles. I propose to amend the article.
2) "Figure 2b, Figure 5a. Y-axis - fonts not recognized. Please fix it.
3) Figure 4 analyzes a bulk sample using SEM and EDX, and in Figure 3 analyze "nano samples" using TEM. I suggest the authors for nano samples" to perform an EDX analysis. Then it will be possible to compare the elemental composition of the samples: bulk sample and nano samples.
These comments do not affect the essence of the study and in no way reduce the quality of the work. I recommend publishing the article.
